# MicroRNA Profiling of PRELI-Modulated Exosomes and Effects on Hepatic Cancer Stem Cells

**DOI:** 10.3390/ijms252413299

**Published:** 2024-12-11

**Authors:** Boyong Kim

**Affiliations:** EVERBIO, 131, Jukhyeon-gil, Gwanghyewon-myeon, Jincheon-gun 27809, Republic of Korea; erythro74@korea.ac.kr; Tel.: +82-10-9105-1435

**Keywords:** non-coding RNA, sorafenib, liver cancer stem cell, exosome, PRELI, hepatocyte

## Abstract

The increasing incidence and mortality rates of liver cancer have heightened the demand for the development of effective anticancer drugs with minimal side effects. In this study, the roles of exosomes derived from liver cancer stem cells (LCSCs) with PRELI (Protein of Relevant Evolutionary and Lymphoid Interest) modulation and their miRNAs were investigated to explore their therapeutic properties for liver cancer. Various techniques, such as miRNA profiling, microRNA transfection, overexpression, flow cytometry, Western blotting, and immunocytochemistry, were used to evaluate the effects of exosomes under PRELI up- and downregulation. Downregulated PRELI cellular exosomes (DPEs) reduced the levels of five markers—CD133, CD90, CD24, CD13, and EpCAM—in LCSCs, with the exception of OV-6. Conversely, upregulated PRELI cellular exosomes (UPEs) significantly increased the expression of CD90, CD24, and CD133 in NHs, with the maximum increase in CD24. PRELI upregulation altered expression levels of miRNAs, including hsa-miR-378a-3p (involved in stem-like properties), hsa-miR-25-3p (contributing to cell proliferation), and hsa-miR-423-3p (driving invasiveness). Exosomes with downregulated PRELI inhibited the AKT/mTORC1 signaling pathway, whereas LCSCs transfected with the candidate miRNAs activated it. Additionally, under PRELI upregulation, exosomes showed increased surface marker expression, promoting cancer progression. The modulation of PRELI in LCSCs affected miRNA expression significantly, revealing candidate miRNA targets for liver cancer treatment. Exosomes with PRELI downregulation show potential as a novel therapeutic strategy. Consequently, this study proposes the potential of PRELI-induced exosomes and the three miRNAs as a liver anticancer therapeutic candidate.

## 1. Introduction

The Protein of Relevant Evolutionary and Lymphoid Interest (PRELI) is characterized by its distinct expression patterns and evolutionary conservation. It plays critical roles in various cellular processes, particularly immune system function and stress responses (relating to oxidative stress and mitochondrial function). The PRELI is involved in lymphoid tissue development and the differentiation of immune cells. Its expression is closely regulated during lymphocyte development, reflecting its crucial role in shaping immune cell functionality and maintaining immune homeostasis [1,2,3]. The evolutionary conservation of the protein highlights its fundamental role in preserving cellular integrity and adaptability [1,2,3]. In hepatic cancer, the PRELI interacts with cancer stem cells. Exosomes, small extracellular vesicles, can carry the PRELI and other biomolecules, influencing communication within the tumor microenvironment. Understanding how PRELI-containing exosomes affect liver cancer stem cells (LCSCs) is crucial for uncovering mechanisms underlying tumor progression and developing therapeutic strategies [1,4]. 

LCSCs are a subpopulation of cells within liver tumors that possess self-renewal and differentiation ability and drive tumorigenesis. These cells contribute to tumor initiation, progression, metastasis, and resistance to conventional therapies. LCSCs are characterized by specific surface markers, such as CD133, CD90, the epithelial cell adhesion molecule (EpCAM), CD24, and CD13, which distinguish them from the bulk of differentiated cancer cells [5]. CD133, CD24, and EpCAM activate Wnt/β-catenin; CD90 and CD13 are associated with Notch and JAK/STAT3 and PI3K/AKT and MAPK pathways, respectively [6,7,8,9,10]. LCSCs play a crucial role in hepatic cancer progression due to their ability to undergo asymmetric division, resulting in the generation of new stem cells and differentiated cancer cells. This hierarchical organization contributes to the heterogeneity of liver cancers, allowing LCSCs to adapt to various microenvironments and resist treatment. The AKT signaling pathway plays a critical role in maintaining the self-renewal, survival, and therapeutic resistance of liver cancer stem cells (LCSCs) [11]. The dysregulation of AKT signaling, often through PTEN loss or PIK3CA mutations, promotes LCSC proliferation, metastasis, and the resistance to apoptosis [12]. Targeting AKT has shown promise in reducing LCSC populations, enhancing the efficacy of standard therapies like sorafenib, and suppressing tumor growth and angiogenesis [11]. Combination therapies integrating AKT inhibitors with immunotherapy or targeted agents hold potential for improving HCC treatment by effectively targeting LCSCs. Further research is essential to optimize these strategies for clinical application [13]. The differentiation process is tightly regulated by key signaling pathways, including Wnt/β-catenin, Notch, Hedgehog, and transforming growth factor beta (TGF-β) pathways, which are frequently dysregulated in liver cancers [14]. Targeting LCSCs offers a promising approach to improving treatment outcomes in patients with liver cancer. Strategies include the inhibition of key signaling pathways, the use of immune-based therapies, and the development of drugs that specifically target LCSCs or their microenvironment. Current research is focused on overcoming the challenges of LCSC plasticity and resistance to ensure effective targeting without harming normal stem cells in the liver. Recent advances in understanding LCSCs have led to the development of novel therapeutic approaches, such as differentiation therapy, which aims to drive LCSCs into a more differentiated state, rendering them more susceptible to conventional treatments [15]. 

Exosomes are small extracellular vesicles secreted by various cells, including cancer cells, with key roles in intercellular communication by transferring proteins, RNA, and microRNAs (miRNAs). In cancer, exosomes contribute to tumor growth, metastasis, and drug resistance, making them both a challenge and an opportunity in cancer therapy. Leveraging their natural role in communication, exosomes have emerged as promising therapeutic tools, serving as drug delivery vehicles or as direct therapeutic agents [16]. Exosomes can be engineered to deliver therapeutic agents, such as drugs or siRNAs, directly to cancer cells, enhancing targeting specificity and reducing side effects. Their ability to cross biological barriers and biocompatibility make them ideal for precision therapy. Additionally, miRNA-loaded exosomes can reprogram cancer cells, thereby inhibiting tumor growth and improving responses to existing treatments [17]. Despite their potential, standardizing production, ensuring safety, and scaling up remain challenging and are active areas of research. Nonetheless, exosome-based therapies offer a novel approach to personalized cancer treatment [18].

In this study, I investigated the functions of exosomes derived from PRELI-regulated LCSCs in liver cancer. I performed comprehensive miRNA profiling of these exosomes to elucidate their potential roles. My findings provide a basis for the development of novel biological pharmaceutical materials targeting liver cancer.

## 2. Results

### 2.1. Characteristics of Exosomes from PRELI-Modulated LCSCs

Under the modulation of PRELI levels, LCSCs secreted exosomes with altered PRELI contents (Figure 1). Following PRELI overexpression and knockdown, I isolated UPEs and DPEs from LCSCs (Figure 1). Compared with the levels in CEs, PRELI levels in UPEs were upregulated approximately 2.37-fold and were downregulated in DPEs 0.62-fold (Figure 1a). Cytotoxicity assays revealed that the effective treatment concentrations were 30 μg/mL for normal hepatocytes and 40 μg/mL for LCSCs (Appendix A). 

The modulation of the PRELI in LCSCs affected the composition of miRNAs in exosomes significantly, in addition to changes in the levels of PRELI (Figure 2). As shown in the heat map, 41 miRNAs exhibited altered levels in the induced exosomes (Figure 2a). Notably, 18 miRNAs showed significant alterations in exosomes derived from LCSCs under UPEs and DPEs (Figure 2a). These 18 miRNAs were associated with 13 biochemical categories, and seven genes were associated with 6 categories (4, 5, 6, 8, 10, and 13) (Figure 2b). Compared with that of the upregulated PRELI, the downregulated PRELI had a more pronounced impact on miRNA levels in exosomes (Figure 2).

The miRNA profiling revealed that miRNAs in six out of all categories were significantly altered under the downregulated PRELI, identifying candidate miRNAs associated with the modulation of the PRELI (Figure 3). Notably, hsa-miR-378c showed the greatest difference between UPEs and DPEs, with a fold change value of 6 (Figure 3). Furthermore, this miRNA type was only associated with the cell cycle category (Figure 3). Additionally, hsa-miR-378a-3p, hsa-miR-25-3p, hsa-miR-423-3p, and hsa-miR-7f-5p in UPEs were associated with all categories (Figure 3).

### 2.2. Effects of Exosomes Derived from PRELI-Modulated LCSCs

The modulation of the PRELI affected the expression of various markers on LCSCs and NHs. Under UPEs, LCSCs exhibited dramatically upregulated levels of various markers (Figure 4), including CD133, CD90, and CD24, with a pronounced increase in CD24 expression (Figure 4). In contrast, UPE had minimal influence on the expression of CD13, EpCAM, or OV-6 in LCSCs (Figure 4). 

Notably, DPEs downregulated the remaining five markers, except for OV-6, in LCSCs, with pronounced reductions in CD133, CD90, CD24, CD13, and EpCAM levels (Figure 4). Interestingly, UPEs induced NHs to adopt characteristics of LCSCs (Figure 5). UPEs upregulated CD90, CD24, and CD133 significantly in NHs, with the most substantial increase noted for CD24 (Figure 5). Additionally, UPEs upregulated the remaining markers in NHs, except for EpCAM (Figure 5). Although DPEs suppressed the expression of CD133, CD90, and CD24 in NHs, the effects were relatively weak (Figure 5).

UPEs activated the AKT signaling pathway, a central pathway in LCSCs that regulates survival, therapeutic resistance, metastasis, and metabolic reprogramming in both LCSCs and NHs. In contrast, DPEs suppressed the AKT signaling pathway (Figure 6). The effects of UPEs on AKT signaling were greater in NHs than in LCSCs. Likewise, DPEs exerted a more pronounced suppressive effect on AKT signaling in NHs (Figure 6). Notably, although DPEs suppressed AKT signaling to approximately 0.32 times the maximum level, UPEs activated AKT signaling to approximately 3.1 times that in NHs. 

### 2.3. Effects of the miRNA Candidates on LCSCs

I identified three differentially expressed miRNA candidates associated with six functional categories and evaluated their regulatory effects on AKT signaling (Figure 7). The expression of AKT signaling molecules was activated in LCSCs upon transfection with the three candidates (Figure 7). Notably, hsa-miR-378a-3p exhibited the greatest effect, resulting in AKT signaling levels approximately 1.72 times greater than those of the control (Figure 7). 

The AKT signaling pathway is a central regulator of drug resistance in LCSCs. The impact of three miRNAs (hsa-miR378a-3p, hsa-miR25-3p, and hsa-miR423-3) affecting the AKT signaling pathway was evaluated by examining their influence on resistance to sorafenib.

In the results of modeling showed sorafenib-resistant hepatocytes (SRHs) (Figure 8), drug resistance, and PRELI expression levels were increased approximately 1.52 times and 2.53 times, respectively (Figure 8a). When treated with siRNAs targeting the three miRNAs (hsa-miR378a-3p, hsa-miR25-3p, and hsa-miR423-3p), the cellular viability of SRH transfection with three anti-miRNA siRNAs against sorafenib was significantly attenuated by approximately 37% (Figure 8b).

## 3. Discussion

The biological functions of exosomes derived from LCSCs under PRELI modulation were evaluated. The primary aim of this research was to determine the potential clinical value of DPEs and three associated miRNAs (hsa-miR378a-3p, hsa-miR25-3p, and hsa-miR423-3p) as pharmaceutical biomaterials for the treatment and prevention of liver cancer. 

Recent research has shown that the suppression of the PRELI in hepatocytes leads to significant changes in signaling pathways and expression patterns [19]. Specifically, the p53, Hippo, and Notch pathways are involved in liver cell regulation, repair, and carcinogenesis. For example, p53 activation has been linked to inflammatory responses and apoptosis in hepatocytes, which can paradoxically promote liver carcinogenesis [19,20]. Additionally, the Hippo pathway plays a critical role in liver regeneration and repair, particularly in damaged livers where hepatocyte proliferation is impaired [21]. This suppression also affects hepatocyte signaling pathways, like hepatocyte growth factor (HGF), involved in proliferation and survival during liver injury [20,21]. Furthermore, the upregulation of the PRELI activates resistance to anti-cancer drugs in hepatocarcinoma cells [1]. 

The upregulation of the PRELI in exosomes from hepatocytes suggests that PRELI packaged into exosomes participates in intercellular communication, especially during stress or injury conditions. Additionally, it may have roles in mitochondrial stress responses, cellular protection, or survival under adverse conditions, such as liver injury or cancer. PRELI upregulation in these exosomes may reflect an adaptive response to stress in hepatocytes, potentially influencing mitochondrial dynamics, apoptosis, or survival pathways, including p53, Notch, or Hippo pathways, in recipient cells. In this study, the PRELI was modified in exosomes (Figure 1), resulting in alterations in miRNA profiles in the induced exosomes from LCSCs (Figure 2). Among several significantly altered miRNAs, hsa-miR378a-3p, hsa-miR25-3p, and hsa-miR423-3p are related to all major functional categories (Figure 2 and Figure 3). hsa-miR378a-3p is involved in regulating the stem-like properties of LCSCs. It can promote self-renewal and maintain the undifferentiated state of these cells [22]. Additionally, this gene helps LCSCs evade apoptosis, contributing to tumor persistence and progression [22]. hsa-miR25-3p enhances cell proliferation, inhibits cell death, downregulates tumor-suppressor genes, and promotes cell growth in LCSCs [23]. hsa-miR423-3p enhances the invasive ability of LCSCs, contributing to metastasis, and regulates the stemness and aggressive behavior of cancer cells [24]. These results suggest that the upregulation of the PRELI in hepatocytes and LCSCs plays crucial roles in the carcinogenesis of normal hepatocytes and promotes cancer-related processes. The three exosomal miRNAs induced by PRELI play vital roles in these processes during hepatocarcinogenesis. Although modulating PRELI expression is a therapeutic strategy against liver cancers, the three miRNAs may serve as more effective targets for the prevention and treatment of liver cancers.

LCSCs express various markers, including CD13, CD24, CD90, CD133, EpCAM, and oval cell marker 6 (OV-6), on their surfaces [25]. CD24, CD90, and CD133 are associated with the AKT/mTOR signaling pathway, CD13 and 133 are ERK signaling pathway components, and OV-6 and EpCAM are Wnt/β-catenin signaling pathway components [25]. Additionally, CD24, CD133, and EpCAM activate signaling pathways associated with resistance to anti-hepatocarcinoma drugs, including sorafenib, cisplatin, doxorubicin, and fluorouracil (5-FU) [26,27,28]. Different from DPEs, under UPEs, LCSCs and NHs showed the upregulation of CD24, CD90, and CD133 on their surfaces (Figure 4 and Figure 5). According to recent reports [29,30,31], CD24 is involved in the regulation of stem cell properties, enhancing cell adhesion and migration as well as tumor growth. CD90 and CD133 are associated with tumorigenic potential and invasiveness, respectively [29,30,31]. These findings suggest that UPEs enhance the sensitivity of cells to external stimuli and promote cancer activity by upregulating surface marker expression. Moreover, these effects of UPEs facilitate the initiation of hepatocarcinogenesis from normal hepatocytes. Interestingly, DPEs or anti-miRNAs targeting these miRNAs hold potential as therapeutic agents for liver cancers. 

One of the key findings of this study was the identification of an additional signaling pathway related to the treatment of liver cancer. Although p53, Notch, and Hippo pathways have provided a basis for the development of various liver cancer drugs [32,33], liver cancer drugs (sorafenib) targeting these pathways often show significant side effects, including hand–foot syndrome, hypertension, diarrhea, and fatigue [34]. Although DPEs suppressed the AKT/mTORC1 signaling pathway in LCSCs and LH (Figure 6), the transfection of the three miRNAs identified in this study in LCSCs activated the signaling pathway (Figure 7). Interestingly, SRH cells transfected with the anti-miRNA siRNAs showed attenuated resistance to sorafenib, similar to the resistance levels observed in NH cells (Figure 8). The AKT signaling pathway was found to modulate resistance to sorafenib [35]. These results suggest that the three anti-miRNA siRNAs have potential as therapeutic agents to prevent drug resistance in liver cancer treatment. A more detailed analysis of how PTEN and CTNNB1 mutations influence liver cancer is needed. Insights from WNT/Lgr5 studies could be integrated, as cancer stem cells (CSCs) drive tumor initiation, growth, and metastasis, playing crucial roles in drug resistance. Specifically, Lgr5 has been highlighted as a novel biomarker in various human cancers, promoting CSC proliferation and self-renewal via the Wnt/β-catenin pathway. Targeting Lgr5+ CSCs shows promise, but single-target therapies might be insufficient for complete eradication and recurrence prevention. Drug screening for Lgr5-related pathways and antibody or drug delivery strategies targeting Lgr5 could be relevant [36,37]. Whether PRELI-regulated exosomes affect CSC function should also be discussed to provide a more comprehensive understanding.

However, as this research primarily relies on in vitro models, further validation in in vivo models is required to elucidate the mechanisms and optimize miRNA delivery methods for clinical applications, presenting a new direction for innovative liver cancer therapies. Furthermore, the mechanisms underlying the interaction between the PRELI, exosomal miRNAs, and the AKT/mTORC1 pathway require further clarification, especially using in vivo models. The data in this study are largely based on in vitro experiments, which may not fully represent the complexity of liver cancer progression and treatment responses. Additionally, while the candidate miRNAs show potential for therapeutic applications, their specificity and potential off-target effects need to be thoroughly evaluated. Future research should also focus on optimizing the delivery mechanisms for these miRNAs to ensure their stability and efficacy in clinical settings. Addressing these limitations will be crucial for translating these findings into viable therapeutic strategies for liver cancer.

Consequently, both DPEs and UPEs altered the expression of various miRNAs, influencing cancer-related processes in LCSCs, promoting tumorigenesis in NHs, and modulating the AKT/mTORC1 signaling pathway. Notably, three specific miRNAs (hsa-miR-378a-3p, hsa-miR-25-3p, and hsa-miR-423-3p) played a pivotal role in regulating the AKT/mTORC1 signaling pathway in LCSCs. The combined action of DPEs and these miRNAs represents a promising avenue for the development of new therapeutic strategies for liver cancers. 

## 4. Materials and Methods

### 4.1. Cell Culture and Establishment of Dosages

Liver cancer stem cells (LCSCs) (sku: 36116-43; Celprogen, Torrance, CA, USA) and normal hepatocytes (NHs) (THLE-3; ATCC, Manassas, VA, USA) were cultured with Human Liver Cancer Stem Cell Media (Celprogen) and BEGM (Bronchial Epithelial Cell Growth Medium) (Lonza, Workingham, UK) using BEGM Bullet Kits (Lonza) at 37 °C, 5% CO_2_. The cultured LCSCs and NHs were exposed to induced exosomes for one day to establish exosomal treatment dosages. To evaluate viability, all exposed cells were stained with Annexin V-conjugated propidium iodide (PI) (Invitrogen, Carlsbad, CA, USA) and analyzed using a flow cytometer (FACSCalibur, BD Biosciences, San Jose, CA, USA) and FlowJo 10.10 software (BD Biosciences) (Appendix A).

### 4.2. PRELI Knockdown and Overexpression

For PRELI knockdown, cultured LCSCs were exposure to Lipofectamine 2000 reagent (Invitrogen, Waltham, MA, USA) with PRELI siRNA or control siRNA oligonucleotides (negative and positive) (Bioneer, Daejeon, Korea) for 48 h. For PRELI overexpression, the cultured LCSCs were exposed to PRELI Lentiviral Activation Particles (h): sc-411975-LAC (Santa Cruz Biotechnology, Houston, TX, USA) with RetroNectin (Takara, Tokyo, Japan) for 48 h. 

### 4.3. Purification of Induced Exosomes and microRNA Profiling

To isolate induced exosomes, the supernatants were collected from LCSCs under PRELI knockdown and overexpression conditions. The control exosomes (CEs), upregulated PRELI exosomes (UPEs), and downregulated PRELI exosomes (DPEs) were isolated and purified from the supernatants (10 mL) using an exoEasy Maxi Kit (QIAGEN, Hilden, Germany) and CD68 Exo Flow Capture Kit (System Biosciences, Palo Alto, CA, USA). The concentrations of isolated exosomes were evaluated using an Exosome Standards Kit (SAE0193; Sigma-Aldrich, St. Louis, MO, USA). The isolated and purified exosomes were sequenced by ebiogen Inc. (Seoul, Republic of Korea) to analyze exosomal functions. An Agilent 2100 Bio-analyzer and RNA 6000PicoChip (Agilent Technologies, Amstelveen, The Netherlands) were used to evaluate RNA quality. RNA was quantified using a NanoDrop 2000 spectrophotometer (Thermo Fisher Scientific, Waltham, MA, USA). Small RNA libraries were prepared and sequenced using an Agilent 2100 Bio-analyzer instrument for a high-sensitivity DNA assay (Agilent Technologies, Inc., Santa Clara, CA, USA), and the NextSeq500system was used for single-end 75 bp sequencing (Illumina, San Diego, CA, USA). To obtain an alignment file, the sequences were mapped using Bowtie 2 (CGE Risk, Lange Vijverberg, The Netherlands), and read counts were extracted from the alignment file using bedtools (v2.25.0) (GitHub, Inc., San Francisco, CA, USA) and R language (version 3.2.2) (R studio, Boston, MA, USA) to evaluate miRNA expression levels based on hg38 genome. miRWalk 2.0 (Ruprecht-Karls-Universität Heidelberg, Medizinische Fakultät Mannheim, Germany) was used for the miRNA target analysis, and ExDEGA v.2.0 (ebiogen Inc., Seoul, Republic of Korea) was used to generate radar charts.

### 4.4. Western Blotting and Exosomal Images

Total proteins were extracted from the induced exosomes (CEs, UPEs, and DPEs) using a DC Protein Assay Kit (BIO-RAD, Hercules, CA, USA). The lysed total protein was electrophoresed using the KOMA EzWay™ PAG System (Bellingham, WA, USA), followed by blotting on a nitrocellulose membrane, blocking for 1 h with 5% skim milk (Sigma) in phosphate-buffered saline with Tween 20 (PBST; Sigma), and washing with PBST. The blots were then reacted overnight with anti-CD-63 (Abcam, Cambridge, UK), anti-PRELI (sc-100817, Santa Cruz Biotechnology), and FITC-anti-CD-63 (Abcam, Cambridge, UK) antibodies diluted to 1:10,000 at 4 °C. Next, samples treated with anti-CD63 were washed with PBST and treated with HRP-conjugated anti-rabbit and anti-mouse IgG (Sigma) diluted by 1:5000. The blots were treated with ECL Reagent (Sigma) and analyzed using iBright FL1000, iBright Analysis 4.0.0 (Invitrogen, Waltham, MA, USA), and Prism 7 (GraphPad, San Diego, CA, USA). 

### 4.5. Evaluation of Exosomal Purification and Transfection

The isolated exosomes were stained with FITC-anti-CD-63 and evaluated using a flow cytometer (BD FACSCalibur, BD Biosciences), FlowJo 10.6.1 (BD Biosciences), a fluorescence microscope (Eclipse Ts-2, Nikon, Shingawa, Japan), the imaging software NIS-elements V5.11 (Nikon), and Prism 7 (GraphPad). To evaluate transfection, LCSCs were exposed to FITC-labeled miRNAs (hsa-miR-378a-3p; Cy5-5′-UCUGGCUCAGGUGGCAUUGGAG, hsa-miR-25-3p; Cy5-5′-CAUUGCACUUGUCUCGGUCUGA, hsa-miR-423-3p; Cy5-5′-AGCUCGGUCUGAGGCCCCUCAG) (BIONEER, Daejeon, Republic of Korea) for one day. During incubation, LCSCs were transfected with miRNAs using Lipofectamine 2000 reagent (Invitrogen) with control siRNA oligonucleotide (negative) (Bioneer) and GFP-GAPDH siRNA (Bioneer) for one day. The treated LCSCs were evaluated for transfection efficiency using a fluorescence microscope (Eclipse Ts-2, Nikon) and NIS-elements V5.11 (Nikon). 

### 4.6. Evaluation of Marker Expression and ATK/mTORC1 Signaling

After exposure to CEs, UPEs, and DPEs with the labeled miRNAs for one day, markers (CD13, 24, 90, 133, EpCAM, and OV-6) on LCSCs and NHs were stained with FITC-anti-CD 133 (Abcam, Cambridge, UK), FITC-anti-EpCAM (Abcam), FITC-anti-CD90 (Abcam), FITC-anti-CD24 (Abcam), FITC-anti-CD13 (Abcam), and FITC-anti-OV-6 (Novus Biologicals, Centennial, CO, USA) for 3 d at 37 °C. Similarly, the cells treated with the labeled miRNAs for one day were stained with these antibodies for 3 d at 37 °C. Expression levels in stained cells were evaluated using a flow cytometer (BD FACSCalibur), FlowJo 10.6.1 (BD Biosciences), and Prism 7 (GraphPad). 

### 4.7. Modeling of Sorafenib-Resistant Hepatocytes and Blocking of Candidate microRNAs

NHs were cultured with sorafenib (Sigma) at serial concentrations of 0.005, 0.007, 0.010, and 0.015 μmol/L for 70 d to model drug-resistant cells. After the exposure, the cellular viability and the expression levels of PRELI were evaluated using Annexin V-PI (Invitrogen) and immunocytochemistry with FITC-anti-PRELI (sc-100817, Santa Cruz Biotechnology), respectively, and were analyzed using a flow cytometer (BDFACScalibur, BD Biosciences), and FlowJo 10.6.1 (BD Biosciences). To suppress the miRNA candidates, NHs were exposed to the synthesized siRNA against hsa-miR-378a-3p, hsa-miR-25-3p, and hsa-miR-423-3p; FITC-3′-AGACCGAGUCCACCGUAACCUC, FITC-3′-GUAACGUGAACAGCCAGACU, and FITC-3′-UCGAGCAGACUCCGGGGAGUC during 48 h. The transfected cells were exposed to sorafenib, and their cellular viability was evaluated using Annexin V-PI (Invitrogen), a flow cytometer (BDFACScalibur, BD Biosciences), and FlowJo 10.6.1 (BD Biosciences).

### 4.8. Statistical Analysis

To establish quantitative and correlative data, an analysis was performed using a one-way analysis of variance (ANOVA) and post hoc Scheffe’s tests using Prism 7 (GraphPad).

## 5. Conclusions

This study demonstrates that PRELI upregulation in hepatocyte- and LCSC-derived exosomes play a pivotal role in liver cancer progression by altering miRNA profiles, particularly hsa-miR-378a-3p, hsa-miR-25-3p, and hsa-miR-423-3p. These miRNAs regulate stem-like properties, proliferation, invasiveness, and apoptosis evasion through the AKT/mTORC1 signaling pathway.

Furthermore, the UPE-induced upregulation of cancer-associated markers (CD24, CD90, and CD133) promotes tumorigenesis and drug resistance. Anti-miRNA siRNAs targeting these miRNAs effectively reduced sorafenib resistance, presenting a promising therapeutic approach. These findings provide novel strategies for liver cancer treatment, focusing on miRNA modulation and PRELI-regulated pathways.

## Figures and Tables

**Figure 1 ijms-25-13299-f001:**
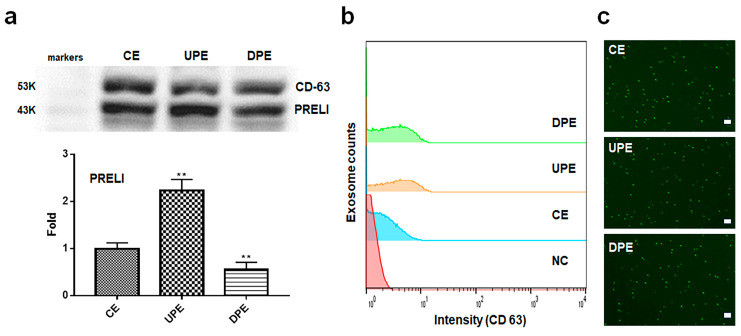
Purification of exosomes isolated from PRELI-modulated LCSCs. (**a**) Marker detection (CD63) and expression levels of PRELI in purified exosomes, as evaluated using Western blotting. Evaluation of purified exosomes with FITC-CD63 antibodies using flow cytometry (**b**) and fluorescence microscopy (**c**). CE: control cellular exosome, UPE: upregulated PRELI cellular exosome, DPE: downregulated PRELI cellular exosome, NC: unstained exosome (** *p* < 0.01), scale bar = 20 µm.

**Figure 2 ijms-25-13299-f002:**
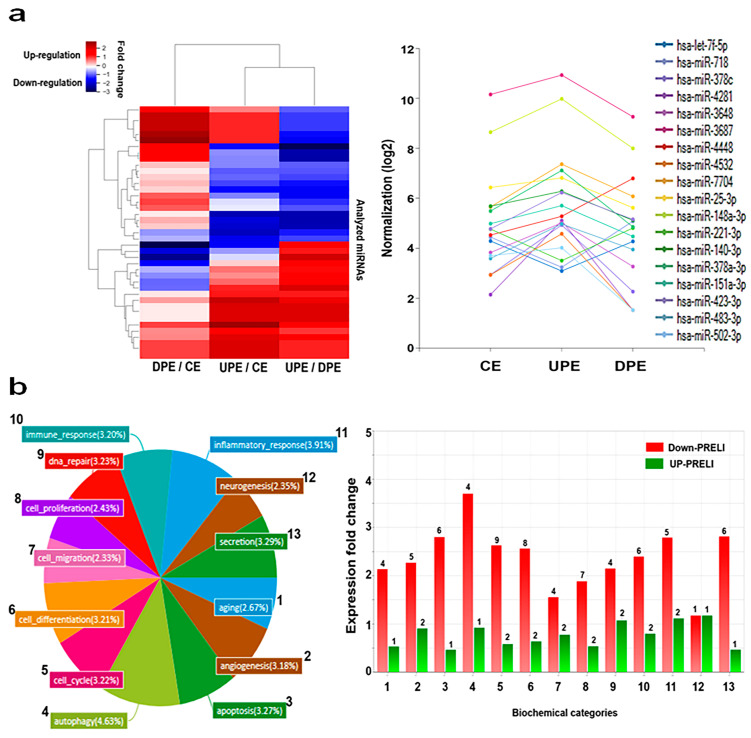
Profiling of exosomes from PRELI-modulated LCSCs. (**a**) Analyzed miRNAs (heat map) and identification of significant miRNAs in the exosomes. (**b**) Comparison of miRNA levels among three types of exosomes. Biochemical categories of the pie chart (1. aging, 2. angiogenesis, 3. apoptosis, 4. autophagy, 5. cell cycle, 6. cell differentiation, 7. cell migration, 8. cell proliferation, 9. DNA repair, 10. immune response, 11. inflammatory response, 12. neurogeneration, 13. secretion) associated with significant miRNAs in the induced exosomes and alterations of miRNA levels in each of the functional categories. The numbers above bar graphs indicate the number of differentially expressed miRNAs. CE: control cellular exosome, UPE: upregulated PRELI cellular exosome, DPE: downregulated PRELI cellular exosome.

**Figure 3 ijms-25-13299-f003:**
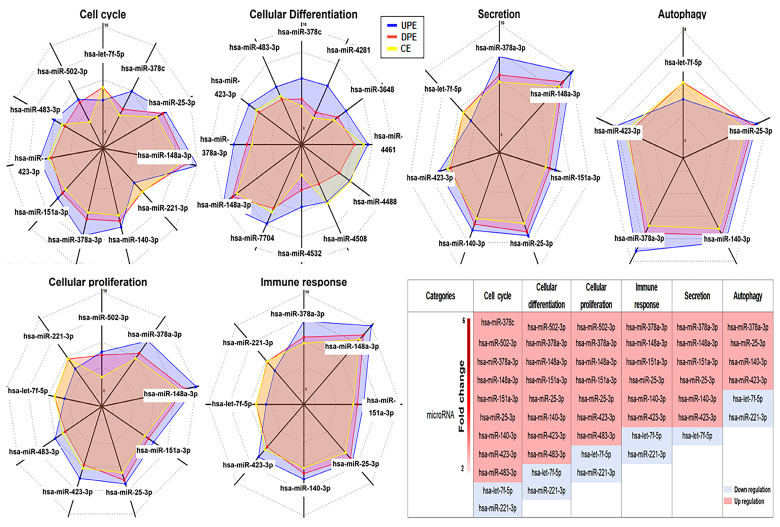
Alterations of miRNAs associated with six biochemical categories in three types of exosomes. (CE: control cellular exosome, UPE: upregulated PRELI cellular exosome, DPE: downregulated PRELI cellular exosome) (*p* < 0.05).

**Figure 4 ijms-25-13299-f004:**
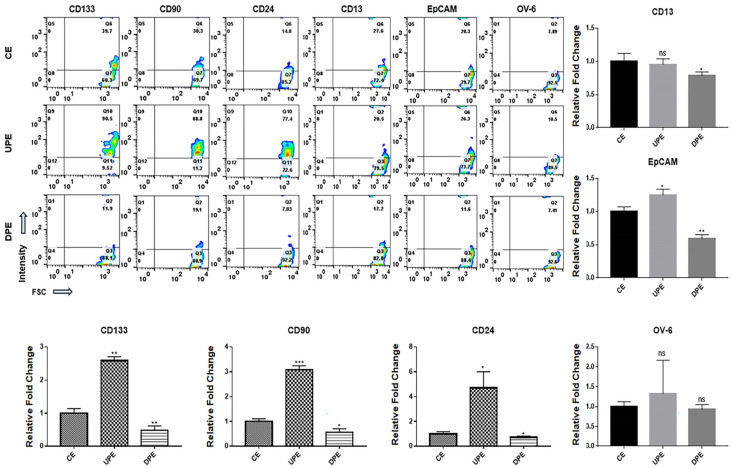
Expression of markers in LCSCs exposed to various exosomes. (CE: control cellular exosomes, UPE: upregulated PRELI cellular exosome, DPE: downregulated PRELI cellular exosome). Fill-patterned bar graphs indicate markers with significant changes. Ns: not significant (* *p* < 0.05, ** *p* < 0.01, *** *p* < 0.001).

**Figure 5 ijms-25-13299-f005:**
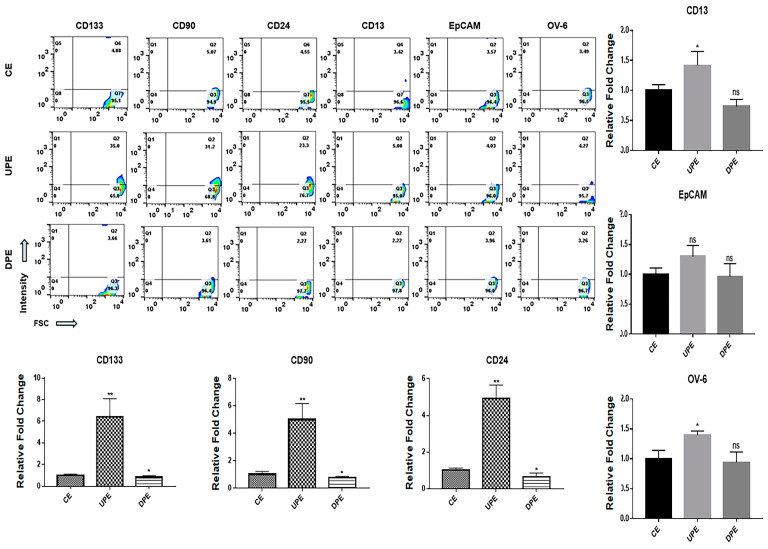
Expression of typical LCSC markers in normal hepatocytes exposed to various exosomes. (CE: control cellular exosome, UPE: upregulated PRELI cellular exosome, DPE: downregulated PRELI cellular exosome). Fill-patterned bar graphs indicate markers with significant changes. Ns: not significant (* *p* < 0.05, ** *p* < 0.01).

**Figure 6 ijms-25-13299-f006:**
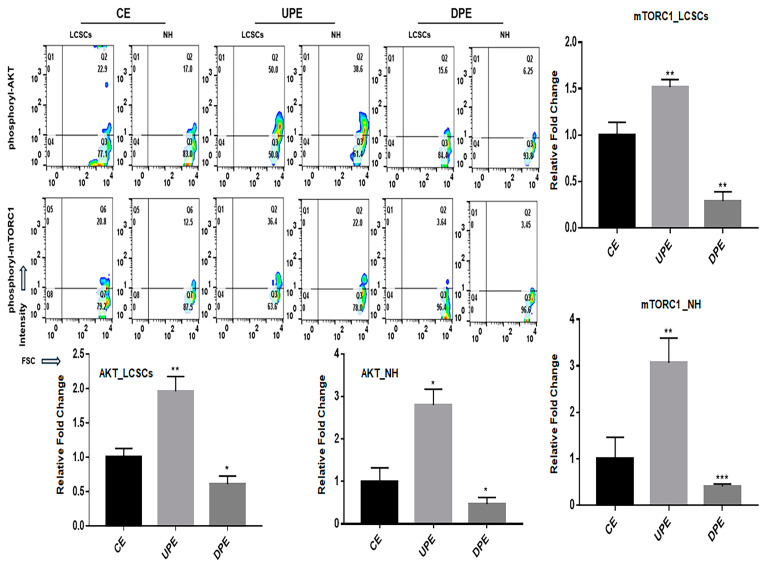
Levels of AKT signaling molecules (phosphorylated AKT and phosphorylated mTORC1) in normal hepatocytes and LCSCs under various exosomes. (CE: control cellular exosome, UPE: upregulated PRELI cellular exosome, DPEs: downregulated PRELI cellular exosome) (* *p* < 0.05, ** *p* < 0.01,*** *p* < 0.001).

**Figure 7 ijms-25-13299-f007:**
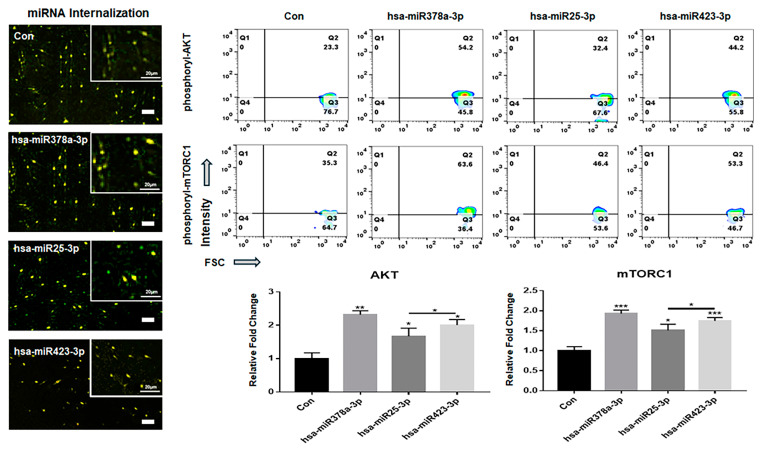
Expression of AKT signaling molecules in LCSCs under miRNA treatment. The merged images show the internalized fluorescence-labeled GAPDH-siRNAs (GFP: green fluorescent proteins) and miRNA (Cy5: cyanine 5) in LCSCs. The expression levels of AKT signaling molecules (phosphorylated AKT and phosphorylated mTORC1) in LCSCs transfected with the fluorescent (Cy5)-labeled miRNAs including hsa-miR378a-3p, hsa-miR25-3p, and hsa-miR423-3p and positive control siRNAs (GFP-GAPDH -siRNA) (* *p* < 0.05, ** *p* < 0.01, *** *p* < 0.001) (scale bars = 20 μm).

**Figure 8 ijms-25-13299-f008:**
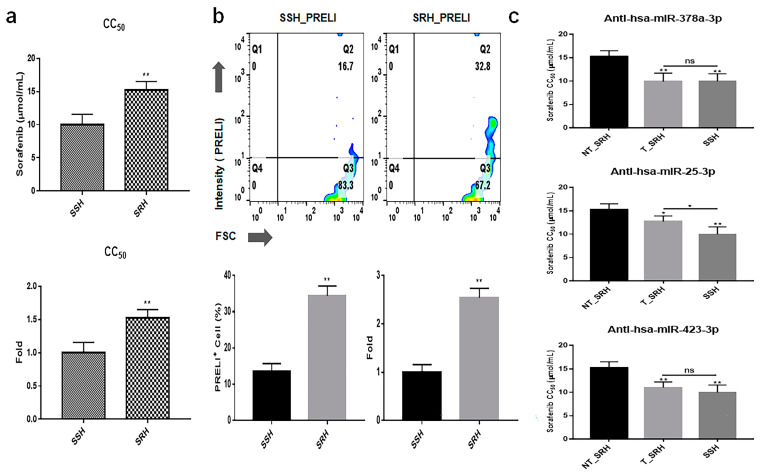
Attenuation of drug resistance and cellular viability in SRHs upon inhibition of three miRNAs. (**a**) Results from modeling drug resistance. (**b**) Expression levels of PRELI in SSHs (sorafenib-sensitive hepatocytes) and SRH (sorafenib-resistant hepatocytes). (**c**) Drug resistance in cells transfected with anti-miRNA siRNAs. Fill-patterned bar graphs indicate sensitivity for Sorafenib.CC_50_: cytotoxic concentration 50; NT: non-transfected; T: transfected, Ns: not significant (* *p* < 0.05, ** *p* < 0.01).

## Data Availability

Data are contained within the article and Appendix A.

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
