# Peer review of "MicroRNA Profiling of PRELI-Modulated Exosomes and Effects on Hepatic Cancer Stem Cells"

_ijms, 2024, doi:10.3390/ijms252413299_

Round 1
Reviewer 1 Report
Comments and Suggestions for Authors
This study explores the role of PRELI (Protein of Relevant Evolutionary and Lymphoid Interest)-regulated exosomes in liver cancer stem cells (LCSCs), focusing on their impact on miRNA expression and therapeutic potential. The findings reveal that upregulation of PRELI enhances the expression of miRNAs such as hsa-miR-378a-3p, hsa-miR-25-3p, and hsa-miR-423-3p within exosomes, promoting LCSC stemness and invasiveness while activating the AKT/mTORC1 signaling pathway. Conversely, PRELI downregulation reduces these miRNA levels, inhibits the AKT pathway, and weakens tumorigenic potential.
The study suggests that exosomes from PRELI-downregulated cells (DPE) and anti-miRNA strategies targeting key miRNAs may reduce liver cancer resistance to drugs like sorafenib and serve as novel therapeutic tools. However, as this research primarily relies on in vitro models, further validation in in vivo models is required to elucidate the mechanisms and optimize miRNA delivery methods for clinical application, presenting a new direction for innovative liver cancer therapies.
Areas for Improvement:
Figures (Lines 393-401, Figures 1-8): The images appear stretched or distorted, which compromises their clarity and presentation. All figures should be revised to correct these issues.
And The words in the picture should be enlarged.
Discussion on PTEN and CTNNB1 Mutations (Line 283): A more detailed analysis of how PTEN and CTNNB1 mutations influence liver cancer is needed. Insights from WNT/Lgr5 studies could be integrated, as cancer stem cells (CSCs) drive tumor initiation, growth, and metastasis, playing crucial roles in drug resistance. Specifically, Lgr5 has been highlighted as a novel biomarker in various human cancers, promoting CSC proliferation and self-renewal via the Wnt/β-catenin pathway. Targeting Lgr5+ CSCs shows promise, but single-target therapies might be insufficient for complete eradication and recurrence prevention. Drug screening for Lgr5-related pathways and antibody or drug delivery strategies targeting Lgr5 could be relevant (Jimin Han et al., IJMS, 2021; Jia He et al., BBRC, 2023). Whether PRELI-regulated exosomes affect CSC function should also be discussed to provide a more comprehensive understanding.
Comments on the Quality of English LanguageThe English could be improved to more clearly express the research.
Author Response
First, I appreciate your comments on our manuscript to improve its quality. We have marked the revised sentences with a red underline based on your comments, and the general comments with a yellow underline
Reviewer 1]
Comment1] The study suggests that exosomes from PRELI-downregulated cells (DPE) and anti-miRNA strategies targeting key miRNAs may reduce liver cancer resistance to drugs like sorafenib and serve as novel therapeutic tools. However, as this research primarily relies on in vitro models, further validation in in vivo models is required to elucidate the mechanisms and optimize miRNA delivery methods for clinical application, presenting a new direction for innovative liver cancer therapies.
Answer 1] I added limitation at the conclusion sections for the mechanisms and optimize miRNA delivery methods for clinical application and in vitro models
Comment2] Figures (Lines 393-401, Figures 1-8): The images appear stretched or distorted, which compromises their clarity and presentation. All figures should be revised to correct these issues.
Answer 2] I revised the figures based on your comment
Comment3] Discussion on PTEN and CTNNB1 Mutations (Line 283): A more detailed analysis of how PTEN and CTNNB1 mutations influence liver cancer is needed. Insights from WNT/Lgr5 studies could be integrated, as cancer stem cells (CSCs) drive tumor initiation, growth, and metastasis, playing crucial roles in drug resistance. Specifically, Lgr5 has been highlighted as a novel biomarker in various human cancers, promoting CSC proliferation and self-renewal via the Wnt/β-catenin pathway. Targeting Lgr5+ CSCs shows promise, but single-target therapies might be insufficient for complete eradication and recurrence prevention. Drug screening for Lgr5-related pathways and antibody or drug delivery strategies targeting Lgr5 could be relevant (Jimin Han et al., IJMS, 2021; Jia He et al., BBRC, 2023). Whether PRELI-regulated exosomes affect CSC function should also be discussed to provide a more comprehensive understanding.
Answer 3] I added you’re the discussion at the Discussion section
Comment4] The quality of English does not limit my understanding of the research.
I received the second English editing service
Reviewer 2 Report
Comments and Suggestions for Authors
An interesting paper on miRNA profiling of exosomes in hepatic cancer. I believe that moderate amendments are required. Please answer or consider the following:
(1) Just to make sure, is the authorship list complete? This is a substantial amount of work for a single researcher, which (if turns out correct), is much appreciated. However, if the authorship is currently complete, I think there should be a change in all sentences where words such as “we” or “our” are used, similar to declarations under the main text (such as “all authors”, “the remaining authors declare” etc.)
(2) I noticed that both “ijms-3323759-original-images” and “ijms-3323759-supplementary” have the same content. In my opinion, original images should have only western blotting images whereas supplementary files should refer to measurements related to exosomes and cytotoxicity. As for the western blotting, if possible, please upload a full membrane with all borders visible and not in grayscale.
(3) Abstract, line 11: is the word “value” or “properties” (or similar) missing after “therapeutic”?
(4) Figures should be enlarged to the borders of each page because currently they are sometimes illegible. Moreover, try to put images of higher resolution and upload full-scale files in the system.
(5) Results, multiple locations: I think that paragraphs should precede respective figures, so move them above figures instead of putting them below figures. It is correct in sections 2-2 and 2-3 but not 2-1.
(6) Results, line 103: does “(S1)” refer to supplementary file S1? If yes, please provide the full name in the text (similar to line 300). Moreover, here I think there should be “S2” because cytotoxicity-related CC50 is shown on the second page of supplementary material (the first one is the concentration of induced exosomes). Please double-check all references to supplementary files in the text.
(7) Figure 2: does the legend in the top-left corner of the heatmap refer to the Z-score? Please add. Moreover, what was the source of the 13 biochemical functions described in the figure’s description? I think it should be described in the methodology. At last, delete “p>0.05” in line 116 because there are no significance marks in this figure.
(8) Results, lines 123-125: could you please elaborate based on what results you concluded that “upregulated PRELI had a more pronounced impact on miRNA levels”? Fold-change seems to be higher in the “Down-PRELI” group (red) in Figure 2b.
(9) Figure 3 is completely illegible, especially radar charts. Enlarging the figures, extending their range to the left of each page, and improving the quality/resolution should help with results presentation in all sections.
(10) Results, line 127: why there are suddenly six biochemical categories instead of thirteen? Please explain in the text; a sentence or two is enough.
(11) Figure 4, barplots: if you present fold-changes, I think it is relative to the specific group, so it must be specified on X-axis, similar to annotations of the heatmap in Figure 2. Double-check other figures/plots.
(12) Similar to my comment no. 10, it is not explained why there is a focus on Akt pathway at the end of section 2-2. Adding a sentence or two will clarify it to the Readers.
(13) It is hard to spot any evident changes in microscopic images in Figure 7 because of their quality. As mentioned earlier, please make sure that high-quality images are uploaded. Moreover, there is a typo in “hsa” – all abbreviations of miRNA have the prefix “has”.
(14) Similar to my comments no. 10 and 12, it is not explained why there is a focus on sorafenib in sections 2-3. I am wondering if your manuscript would benefit from the Results and Discussion section instead of them being separated. This should clarify some study directions on the first mention. Please consider doing it, especially since you refer to figures in Discussion (e.g., lines 240 and 241), although it is not obligatory if you add a sentence or two justifying your decision.
(15) Results, lines 211-216: delete these sentences, they are a part of the manuscript’s template and are probably left accidentally. Please make sure that “3. Discussion” is deleted in line 211 because it is currently a part of the Figure 8 description.
(16) Discussion, line 289: add a full stop at the end of the sentence.
(17) Materials and Methods, mention cell culture conditions in section 4-1.
(18) Materials and Methods, section 4-3: what version of human reference was used in your study? hg19, hg38, T2T-CHM13? If the first one, why an outdated reference was used?
(19) Materials and Methods, sections 4-8: mention the type of data that were subjected to analysis of statistical significance.
(20) Conclusions: In my opinion, they are too long and do not gravitate towards the main findings. Limitations (the second paragraph) should be a part of the Discussion. Try to put short and simple take-home messages in your Conclusions section.
(21) Italicize “in vivo” and “in vitro” whenever used in your manuscript.
(22) Supplementary Materials, lines 414-415: If you intend to put western blotting images as part of supplementary materials, please consider my comment no. 2 and put it as “S3” and not “S2”. I think S1 is related to exosomes and S2 to cytotoxicity.
(23) Acknowledgments, line 423: what kind of support was given by EVERBIO, if the study did not receive any funding (as mentioned in line 419)?
Author Response
First, I appreciate your comments on our manuscript to improve its quality. We have marked the revised sentences with a blue underline based on your comments, and the general comments with a yellow underline
Reviewer 2]
(1) Just to make sure, is the authorship list complete? This is a substantial amount of work for a single researcher, which (if turns out correct), is much appreciated. However, if the authorship is currently complete, I think there should be a change in all sentences where words such as “we” or “our” are used, similar to declarations under the main text (such as “all authors”, “the remaining authors declare” etc.)
Answer] I revised all sentences based on your comments
(2) I noticed that both “ijms-3323759-original-images” and “ijms-3323759-supplementary” have the same content. In my opinion, original images should have only western blotting images whereas supplementary files should refer to measurements related to exosomes and cytotoxicity. As for the western blotting, if possible, please upload a full membrane with all borders visible and not in grayscale.
Answer] I added the full membrane with all borders visible without grayscale at the S1 file
(3) Abstract, line 11: is the word “value” or “properties” (or similar) missing after “therapeutic”?
Answer] I revised the sentence.
(4) Figures should be enlarged to the borders of each page because currently they are sometimes illegible. Moreover, try to put images of higher resolution and upload full-scale files in the system.
Answer] I revised their resolution
(5) Results, multiple locations: I think that paragraphs should precede respective figures, so move them above figures instead of putting them below figures. It is correct in sections 2-2 and 2-3 but not 2-1.
Answer] I moved their locations based on your comment
(6) Results, line 103: does “(S1)” refer to supplementary file S1? If yes, please provide the full name in the text (similar to line 300). Moreover, here I think there should be “S2” because cytotoxicity-related CC50 is shown on the second page of supplementary material (the first one is the concentration of induced exosomes). Please double-check all references to supplementary files in the text.
Answer] I revised all based on your comments
(7) Figure 2: does the legend in the top-left corner of the heatmap refer to the Z-score? Please add. Moreover, what was the source of the 13 biochemical functions described in the figure’s description? I think it should be described in the methodology. At last, delete “p>0.05” in line 116 because there are no significance marks in this figure.
Answer] I revised the legend, and the figure based on your comments
(8) Results, lines 123-125: could you please elaborate based on what results you concluded that “upregulated PRELI had a more pronounced impact on miRNA levels”? Fold-change seems to be higher in the “Down-PRELI” group (red) in Figure 2b.
Answer] I revised the sentence
(9) Figure 3 is completely illegible, especially radar charts. Enlarging the figures, extending their range to the left of each page, and improving the quality/resolution should help with results presentation in all sections.
Answer] I revised all the figures to make them legible
(10) Results, line 127: why there are suddenly six biochemical categories instead of thirteen? Please explain in the text; a sentence or two is enough.
Answer] I revised the sentence with more detail
(11) Figure 4, barplots: if you present fold-changes, I think it is relative to the specific group, so it must be specified on X-axis, similar to annotations of the heatmap in Figure 2. Double-check other figures/plots.
Answer] I revised the figure
(12) Similar to my comment no. 10, it is not explained why there is a focus on Akt pathway at the end of section 2-2. Adding a sentence or two will clarify it to the Readers.
Answer] I revised the sentence with more detail at the introduction and result sections
(13) It is hard to spot any evident changes in microscopic images in Figure 7 because of their quality. As mentioned earlier, please make sure that high-quality images are uploaded. Moreover, there is a typo in “hsa” – all abbreviations of miRNA have the prefix “has”.
Answer] hsa stands for "Homo sapiens," indicating that the miRNA is derived from humans. According to the internationally accepted standard nomenclature, human miRNAs are denoted in the format hsa-miR-XXX.
(14) Similar to my comments no. 10 and 12, it is not explained why there is a focus on sorafenib in sections 2-3. I am wondering if your manuscript would benefit from the Results and Discussion section instead of them being separated. This should clarify some study directions on the first mention. Please consider doing it, especially since you refer to figures in Discussion (e.g., lines 240 and 241), although it is not obligatory if you add a sentence or two justifying your decision.
Answer] I added description for a focus on sorafenib in sections 2-3
(15) Results, lines 211-216: delete these sentences, they are a part of the manuscript’s template and are probably left accidentally. Please make sure that “3. Discussion” is deleted in line 211 because it is currently a part of the Figure 8 description.
Answer] I deleted the lines
(16) Discussion, line 289: add a full stop at the end of the sentence.
Answer] I revised the sentence
(17) Materials and Methods, mention cell culture conditions in section 4-1.
Answer] I added the condition in the section
(18) Materials and Methods, section 4-3: what version of human reference was used in your study? hg19, hg38, T2T-CHM13? If the first one, why an outdated reference was used?
Answer] In this study, the research data were obtained through exosome purification and analysis by a company called ebiogen. Upon inquiry, we were informed that the hg38 genome was used, and this information has been added to the Materials and Methods section.
(19) Materials and Methods, sections 4-8: mention the type of data that were subjected to analysis of statistical significance.
Answer] I described the analysis of statistical significance at the section.
(20) Conclusions: In my opinion, they are too long and do not gravitate towards the main findings. Limitations (the second paragraph) should be a part of the Discussion. Try to put short and simple take-home messages in your Conclusions section.
Answer] I revised the section based on your comment
(21) Italicize “in vivo” and “in vitro” whenever used in your manuscript.
Answer] I revised them
(22) Supplementary Materials, lines 414-415: If you intend to put western blotting images as part of supplementary materials, please consider my comment no. 2 and put it as “S3” and not “S2”. I think S1 is related to exosomes and S2 to cytotoxicity.
Answer] S1 file contains all supplementary data including Exosome concentrations, Treatment dosage data and original image of western blotting
(23) Acknowledgments, line 423: what kind of support was given by EVERBIO, if the study did not receive any funding (as mentioned in line 419)?
Answer] Yes, this study received no external funding.
Round 2
Reviewer 1 Report
Comments and Suggestions for Authors
I agree to publish.I was very satisfied with all the answers, although the picture still looked not very comfortable, but I think it did not affect my judgment and understanding
Author Response
I appreciate your comments to improve my manuscript.
I already uploaded the revised figures with high resolution at the Round 1
Reviewer 2 Report
Comments and Suggestions for Authors
Thank you for considering my suggestions. I have only minor comments (numbering as in the first report):
1) Ok, but change the last “our” in the document – it is in the last sentence of the Introduction.
2-3) Ok.
4) They are better but I cannot entirely assess their quality based on the PDF file, I hope you did include the best possible resolution in the separate files uploaded to the system.
5-10) Ok.
11) I think here you need to double-check annotations on X-axes. They are not annotated as in the heatmap being a part of Figure 2.
12) Ok.
13) I understand what “hsa” stands for – I indicated that you have a typo in this prefix: you have “has” instead of “hsa”. Please correct.
14-23) Ok.
Author Response
1) Ok, but change the last “our” in the document – it is in the last sentence of the Introduction.
Answer] I revised the sentence
4) They are better but I cannot entirely assess their quality based on the PDF file, I hope you did include the best possible resolution in the separate files uploaded to the system.
Answer] From the initial submission of the manuscript, I was required to upload high-resolution image files, so I did so. If it is difficult to confirm, please provide your email address, and I will send them to you via email. The final revised figures were uploaded for resubmission.
11) I think here you need to double-check annotations on X-axes. They are not annotated as in the heatmap being a part of Figure 2.
Answer] I revised the figure and its legend
13) I understand what “hsa” stands for – I indicated that you have a typo in this prefix: you have “has” instead of “hsa”. Please correct.
Answer] I revised the figure 7